# *Solanum aethiopicum* L. from the Basilicata Region Prevents Lipid Absorption, Fat Accumulation, Oxidative Stress, and Inflammation in OA-Treated HepG2 and Caco-2 Cell Lines

**DOI:** 10.3390/plants12152859

**Published:** 2023-08-03

**Authors:** Ludovica Lela, Daniela Russo, Filomena De Biasio, Domenico Gorgoglione, Angela Ostuni, Maria Ponticelli, Luigi Milella

**Affiliations:** 1Department of Science, University of Basilicata, V.le Ateneo Lucano 10, 85100 Potenza, Italy; ludovica.lela@unibas.it (L.L.); daniela.russo@unibas.it (D.R.); angela.ostuni@unibas.it (A.O.); 2Spinoff Bioactiplant s.r.l., Via dell’Ateneo Lucano 10, 85100 Potenza, Italy; 3EVRA S.r.l., Località Galdo, 85044 Lauria, Italy; f.debiasio@evraitalia.it (F.D.B.); cto@evraitalia.it (D.G.)

**Keywords:** *Solanum aethiopicum* L., Lucanian eggplant, in vitro investigations, obesity, fat accumulation, oxidative stress, inflammation, lipid absorption

## Abstract

Obesity is widely associated with intestine barrier impairment, nonalcoholic fatty liver disease (NAFLD) outbreaks, oxidative stress, and inflammation. In a previous investigation, the *Solanum aethiopicum* L. growing in Basilicata Region has demonstrated to have antioxidant activity; hence this investigation was aimed to evaluate for the first time the antilipidemic and anti-inflammatory activity of the Lucanian *S. aethiopicum* L. peel extract in vitro on OA-treated HepG2 and Caco-2 cell lines. It was shown that the extract could reduce lipogenesis by down-regulating SREBP-1c and HMGCR expression and fatty acid *β*-oxidation by up-regulating PPAR*α*, CPT1A, and UCP2 expression. In addition, the *S. aethiopicum* L. peel extract might also improve oxidative stress by reducing endoplasmic reticulum stress and regulating the Nrf2 and Nf-κB molecular pathways. Altogether, these results demonstrated for the first time the possible application of the Lucanian *S. aethiopicum* peel extract for preventing obesity and managing NAFLD.

## 1. Introduction

Obesity is a hotly debated topic in health science due to its epidemic nature and deleterious effects. A recently conducted global survey by the World Health Organisation (WHO) disclosed that 1.9 billion subjects over the age of 18 were overweight; of these, more than 650 million were obese. According to the WHO, the body mass index (BMI) can be used to define the condition of overweight or obesity, which is between 25.0 and 29.9 in an overweight condition and above 30 in an obese condition [1]. Underlying the obesity state onset, there is an imbalance between energy expenditure and intake for a prolonged period of time, leading to fat adipose tissue accumulation throughout the body and then weight gain. The excess of macronutrients also leads to a state of oxidative stress and inflammation, which is not only limited to adipose tissue but also to other peripheral tissues, resulting in a condition known as metabolic syndrome (MetS) [2]. The intestine and intestinal barrier integrity have emerged as key players in the development and progression of MetS. The intestine, indeed, is responsible for the absorption of nutrients, including dietary fats and carbohydrates, so impairments in nutrient absorption and transport mechanisms can contribute to obesity, dyslipidemia, and glucose dysregulation observed in metabolic syndrome [3]. Moreover, another recognized hepatic manifestation of MetS is nonalcoholic fatty liver disease (NAFLD), a disease associated with obesity, dyslipidemia, type 2 diabetes, and hypertension [4]. Treatment of MetS and obesity generally needs long-term measures, including non-surgical, such as dietary modification, and medical practitioners, including drug administration and bariatric surgery. However, apart from these treatment options, plants and their derivatives are also widely used worldwide to induce weight loss. It was indeed demonstrated that plant phenolics lowered weight gain and bloodstream lipids and improved oxidative stress and inflammatory condition [1]. In traditional medicine, eggplant species have mainly been used for tackling diabetes, and the Mayo Clinic and American Diabetes Association have also suggested eggplant utilization as a vegetable usable for degenerative conditions management related to diabetes [5,6]. These healthy properties might be associated with the high content of active molecules since eggplants have been revealed to be a rich source of phenolic compounds, alkaloids, and carotenoids [7,8,9] with known antioxidant, anti-inflammatory, anti-obesogenic, and anti-atherosclerotic properties [7,10]. Among these vegetables belonging to the Solanaceae family, the *Solanum aethiopicum* L. or Scarlet eggplant is a typical eggplant consumed in tropical Africa [10]. In Italy, where the scarlet eggplants have been eaten for several years together with the brinjal eggplant (*Solanum melongena*), *Solanum aethiopicum* L. is only grown in Basilicata Region [11]. In a recent investigation, the *S. aethiopicum* L. from the Basilicata Region showed antioxidant activity due to the presence of different phenolic compounds, including hydroxycinnamic acids (4-*O*-Caffeoylquinic acid and Chlorogenic acid), flavanones (Eriodictyol-7-*O*-glucoside, Naringenin-7-*O*-glucoside, Eriodictyol, and Naringenin) and flavanols (Quercetin-3-*O*-rutinoside, Kaempferol-3-*O*-glucoside, and Kaempferol-3-*O*-rutinoside) [12]. Further, several in vitro and in vivo investigations demonstrated that phenolic compounds reduce fasting and food intake, lipogenesis, inflammation, adipocyte differentiation, and growth and increase lipolysis and fatty acids *β*-oxidation [13]. For this reason, it was decided to test the effect of the Lucanian *S. aethiopicum* peel extract on lipid absorption, lipid accumulation, fatty acid metabolism, oxidative stress, and inflammation on in vitro cells models for the first time.

## 2. Results and Discussion 

### 2.1. S. aethiopicum Improved Lipid Absorption in Caco-2 Cells

The human body needs cholesterol and other lipids for cell growth, hormone production, axon myelination, skin barrier formation, and lung surfactant biogenesis. Lipids could be synthesized endogenously or acquired from the diet and are bundled into lipoprotein particles to be transported through the hydrophilic blood and tissue lymph environment. The generation, uptake, and trafficking of these particles are controlled by a network of signaling processes comprising several enzymes and transporters so that mutations of genes coding for these processes lead to the outbreak of dyslipidemias [14]. One of the cell models generally accepted for studying intestinal lipoprotein production is the colon carcinoma Caco-2 cell line, as they differentiate spontaneously to a small intestinal phenotype [15,16]. Considering this background, the Caco-2 cell lines were selected to evaluate the effect of the Lucanian *S. aethiopicum* in improving intestinal lipid absorption. In particular, to select the appropriate concentrations to be used, the extract cytotoxicity was tested by the MTT assay using different concentrations (5–400 µg/mL) [12]. As shown in Figure 1, *S. aethiopicum* peel extract reported no cytotoxic effect at all doses tested after 24 h of treatment. 

Considering these results, Caco-2 cells were treated for 24 h with oleic acid (OA) (0.3 mM) and the Lucanian *S. aethiopicum* peel extract at a concentration of 200 and 100 µg/mL to assess the expression of enzymes involved in cholesterol synthesis, uptake, and efflux by qRT-PCR. A pivotal role in lipid homeostasis and, in particular, in cholesterol transport is played by several ATP-Binding Cassette (ABC) transporters [17]. Specifically, the ABCA1 transporter is highly expressed in peripheral tissues, the intestine, and the liver; its mutation or reduction in activity is related to the impairment in moving cholesterol and phospholipid from cells to the apolipoproteinA-I (apoA-1). This process constitutes the rate-limiting step in high-density lipoprotein HDL biogenesis with the consequent reduction in circulating HDL levels and the increased risk of atherosclerotic lesion formation [14,18]. On the other hand, ABCG5 with ABCG8 form a heterodimer implied in the excretion of cholesterol and dietary sterols from enterocytes and hepatocytes into the intestinal lumen and bile, respectively. Also, in this case, a reduction in the expression of ABCG5 or G8 leads to a premature atherosclerotic process [14]. In this study, Caco-2 cell lines treatment with *S. aethiopicum* peel extract improved the expression of either ABCA1 or ABCG5, contrasting the effect of OA (Figure 2a,b) known to decrease the efflux mediated by ABCA1 [19]. The data obtained are consistent with the increased HDL levels observed after administering the African *S. aethiopicum* stalks extract to diabetic albino rats [20]. Other important gene transcription regulators implied in lipid uptake and synthesis are the Sterol regulatory element-binding protein (SREBPs). Specifically, SREBP-1c is involved in the transcription regulation of enzymes involved in triglycerides (TG) and fatty acids syntheses, such as acetyl-CoA carboxylase (ACC), fatty acid synthetase (FAS), and stearoyl-CoA desaturase (SCD). During overnutrition, it is possible to assist in increased de novo lipogenesis and glycolysis pathways as a consequence of the activation of SREBP-1c, which was seen to be highly expressed in the jejunum and ileum [17,21]. The levels of this transcription factor are significantly elevated in animal models of obesity and obese patient. However, it was seen that polyunsaturated fatty acids could inhibit SREBP-1c, thus reducing adipose tissue mass overproduction and potential metabolic complications associated with obesity [17,22]. African eggplants like *Solanum melongena* and *S. aethiopicum* have been demonstrated to be rich in polyunsaturated fatty acids [23,24]; for this reason, it was decided to evaluate the effect of the Lucanian *S. aethiopicum* peel extract on SREBP-1c demonstrating its ability in reducing its expression. The extract was indeed able to prevent the upregulation of this transcription factor induced by OA treatment (Figure 2c). In the ileum were also highly expressed the 3-hydroxy-3-methylglutaryl-coenzyme A (HMG-CoA) reductase (the cholesterol biosynthesis rate-limiting enzyme) and the low-density lipoprotein (LDL) receptor (LDLr) (responsible for the removal of the circulating cholesterol-rich LDL). The *S. aethiopicum* peel extract was demonstrated to significantly reduce the expression of HMG-CoA reductase (HMGCR) at levels near to that of the control, thus abrogating the induction mediated by OA (Figure 2d), while no effect was seen on LDLr expression (Figure 2e). Altogether these results provide evidence of the potential role of the Lucanian *S. aethiopicum* peel extract in contrasting the expression of mediators involved in the onset of dyslipidemia. This antilipidemic activity may be related not only to the presence of polyunsaturated fatty acids [23,24] but also to the presence of hydroxycinnamic acids like chlorogenic acid and flavonols like quercetin [12]. Previous investigations showed indeed their ability to inhibit HMGCR by binding this receptor similarly to a known drug used for treating dislipidemia pathologies, simvastatin [25]. Moreover, it was recently shown that intestinal inflammation should be in part responsible for intestinal insulin resistance induction and the linked postprandial dyslipidemia. It was indeed demonstrated that the insulin resistance mediated by tumor necrosis factor α (TNF-α) was responsible for a marked lipoprotein overproduction by the gut [26]. Based on these findings, it was decided to test the *S. aethiopicum* peel extract ability in reducing inflammation, observing that it significantly downregulated the expression of two pro-inflammatory cytokines, TNF-α and interleukin-6 (IL-6) (Figure 2f,g). These anti-inflammatory activity may be related to flavonoids, which are abundant in the Lucanian *S. aethiopicum* peel extract [12] and have been shown to improve inflammatory processes by targeting the TLR4/NF-*κ*B pathway [27]. Specifically naringenin was found to be one of the most abundant Lucanian *S. aethiopicum* peel extract compounds (5047.89 ± 509.23 µg/g dry extract) [12] and demonstrated to play an important role as an anti-inflammatory agent throughout the inhibition of TLR4/NF-*κ*B pathway [28]. Likewise, a flavanol found in the eggplant peel extract, kaempferol-3-*O*-rutinoside (2320.18 ± 245.93 µg/g dry extract) [12], reduced the levels of TNF-α mRNA expression and suppressed its expression in RAW264.7 cells stimulated with LPS by inhibiting the nucleus translocation of NF-*κ*B [29].

### 2.2. S. aethiopicum Reduced Lipid Accumulation and Fatty Acid Metabolism in HepG2 Cells

Apart from the intestine, the liver plays a pivotal role in dyslipidemia development since it is implied in the coordination of lipid metabolism. For this reason, it was decided to test the antilipidemic effect of the Lucanian *S. aethiopicum* peel extract on human hepatoma cell lines (HepG2), a common in vitro model used to evaluate the effect of a drug on the liver as it retains many of the normal human hepatocytes specialized characteristics. Our previous investigation demonstrated the absence of *S. aethiopicum* peel extract toxicity, selecting the concentrations of 200 µg/mL and 100 µg/mL as the best dosage for assessing its activity [12]. Therefore, in this investigation, it was decided to evaluate the effect of the same extract on lipid accumulation and the expression of enzymes involved in cholesterol synthesis, uptake, and efflux by qRT-PCR.

HepG2 cells were exposed to the Lucanian *S. aethiopicum* peel extract and OA for 24 h to evaluate lipid accumulation by Oil red O (ORO) staining. ORO is fat-soluble diazo-dye able to stain cholesterol esters and natural lipids but not biological membranes; hence, thanks to its hydrophobic nature, it moves through the solvent to bind to the lipid droplet [30]. As shown in Figure 3, cells treated with OA showed more lipid droplets than control cells. However, the extract counteracted OA-induced lipid accumulation, bringing back a less intense red color, especially at the highest concentration. 

Considering this reduction in lipid accumulation, it was decided to investigate the mechanism underlying this effect. AMP-activated protein kinase (AMPK) is a key regulatory molecule that modulates hepatic lipid metabolism via regulating the transcription of SREBP and peroxisome proliferator-activated receptor α (PPARα) [31]. The Lucanian *S. aethiopicum* peel extract did not up-regulate the expression of AMPK (Figure 4a); however, it influenced the expression of SREBP-1c and PPAR*α*. Specifically, as seen for Caco-2, in HepG2 cells, the *S. aethiopicum* peel extract might downregulate the expression of SREBP-1, thereby decreasing de novo lipogenesis (Figure 4b). This effect was further promoted by reducing HMGCR expression (Figure 4c) necessary for cholesterol synthesis. Hence, it is possible to speculate that the Lucanian *S. aethiopicum* peel extract prevented lipid biosynthesis by acting on two rate-limiting enzymes, ACC and HMGCR. In fact, it may indirectly affect fatty acid synthesis by preventing the expression of ACC through the downregulation of SREBP-1c and directly the cholesterol synthesis by downregulating the HMGCR. On the other hand, the Lucanian *S. aethiopicum* peel extract increased the expression of PPAR*α* (Figure 4d), responsible for the mitochondrial *β-*oxidation enzymes’ up-regulation [31]. The observed results may be attributable to the presence of 5-*O*-Caffeoylquinic acid in the Lucanian eggplant peel (1639.82 ± 117.50 µg/g dry extract) [12] since in vivo investigation on HFD-fed Sprague–Dawley rats revealed that this compound was able to increase the expression of PPARα and decrease that of SREBP-1c [32]. This is an encouraging result as in the liver, PPAR*α* represents an important mediator of the hepatic hunger response; it was seen that PPAR*α* deficient mice, even under fasting conditions, develop fatty liver. This effect is probably also due to the liver’s inability to promote the carnitine palmitoyl transferase 1A (CPT1A) gene expression during fasting. CPT1A is indeed activated by PPAR*α* and represents the carnitine shuttle rate-limiting step during the oxidation of long-chain lipids into the mitochondria [33]. The Lucanian *S. aethiopicum* peel extract was demonstrated to increase the expression of CPT1A, showing greater activity at 100 µg/mL than at 200 µg/mL (Figure 4e). Furthermore, the extract showed to up-regulate the expression of the uncoupling protein 2 (UCP2) (Figure 4f), whose measurement is a widely used method to assess the entity of fatty acid *β*-oxidation. UCPs are, in fact, part of the family of mitochondrial anion transporters that decouple electron transport from the synthesis of ATP, thus generating heat instead of energy. Therefore, UCPs might be pivotal in energy expenditure and can represent possible genes for treating obesity and related metabolic disturbance [34]. Considering these findings, the Lucanian *S. aethiopicum* peel extract should be further investigated to manage NAFLD since it reduced lipogenesis by down-regulating SREBP-1 and HMGCR and improved fatty acid *β*-oxidation by up-regulating PPAR*α*, CPT1A, and UCP2. This activity might be related to its phytochemical composition; as shown in a previous investigation [12], it was rich in phenolic acids and carotenoids involved in improving lipid homeostasis by regulating the expression of PPAR*α* and SREBP-1 [35,36]. Another important result derived from the expression of LDLr, which is known to be significantly low in hyperglycemic patients since it is related to increased LDL cholesterol levels. In contrast with data obtained in Caco-2 cell lines, in which the extract had no effect in increasing the expression of LDLr, in HepG2, the Lucanian *S. aethiopicum* peel extract up-regulated the expression of this receptor (Figure 4g), thereby suggesting a promising role in reducing LDL cholesterol levels and so the risk of CVD onset [37]. These results are in agreement with the presence of rutin in the Lucanian eggplant peel extract (4807.21 ± 543.00 µg/g dry extract) [12] since this specialized metabolite proved to up-regulate the expression of LDLr in HepG2 cell lines [38]. Finally, no effect was seen for ABCA1 (Figure 4h).

### 2.3. Antioxidant Activity of S. aethiopicum in OA-Treated HepG2 Cells

Increased fatty acids are significantly related to increased reactive oxygen species (ROS) generation and the subsequent establishment of an inflammatory process. Hence, it was decided to test the Lucanian *S. aethiopicum* peel extract (200 and 100 µg/mL) on the expression of mediators involved in oxidative stress and inflammation by in vitro assay on HepG2 cell lines treated with OA for 24 h. HepG2, when treated with OA, showed a higher amount of ROS compared to the control; however, the treatment with the extract reduced OA-induced ROS generation, thus restoring basal conditions (Figure 5).

This effect in reducing oxidative stress might be related to the ability of the Lucanian *S. aethiopicum* peel extract to improve the expression of key antioxidant enzymes (Figure 6a–c), such as superoxide dismutase (SOD2), NADPH-quinone oxidase 1 (NQO1), and catalase (CAT), as demonstrated by the qRt-PCR. The transcription of these cytoprotective enzymes is controlled by the nuclear transcription factor NF-E2-related factor 2 (Nrf2), known to play a pivotal role in regulating the cellular redox status. Under homeostatic conditions, this transcriptional factor is repressed by the link with the Kelch-like ECH-associated protein 1 (Keap1), which acts as a negative regulator. However, upon ROS exposure, Keap1 dissociates from Nrf2 with the consequent translocation of Nrf2 from the cytosol to the nucleus. In this cellular compartment, Nrf2 binds the promoter region of the antioxidant response element (ARE) and induces the expression of the antioxidant gene enzymes [39]. The Lucanian *S. aethiopicum* peel extract showed to increase the expression of Nrf2 (Figure 6d); hence it is possible to suppose that it could induce the production of the detoxification enzymes SOD2, NQO1, and CAT by up-regulating this transcriptional factor. Under stress conditions like that caused by ROS, it is possible to observe the endoplasmic reticulum (ER) stress activation, which was recently identified as a key mediator in promoting inflammation, lipid biosynthesis, and apoptosis leading to liver disorders. A marker of ER stress activation is the molecular chaperone binding immunoglobulin protein (BIP), known to be positively correlated to hepatocytic apoptosis [40]. The Lucanian *S. aethiopicum* peel extract drastically reduced BIP activation by OA (Figure 6e), indicating its possible role in reducing ER stress and liver damage. This activity might be related to the presence of *S. aethiopicum* peel extract of flavanols like quercetin [12], which was demonstrated to inhibit BIP by using molecular docking [40]. However, no effect was seen on C/EBP homologous protein (CHOP) (Figure 6f), another important mediator of ER stress-induced organ injury [41]. ER stress is also related to activating the nuclear factor kappa light chain enhancer of activated B cells (NF-κB), a transcription factor that has an important role in the innate immune response since it is involved in the expression of pro-inflammatory mediators like cytokines [42,43]. The activation of this transcriptional mediator is down-regulated by treating HepG2 with the Lucanian *S. aethiopicum* peel extract (Figure 6g), leading to the reduction of the pro-inflammatory cytokine IL-6 (Figure 6h). Underlying the promising effects of the Lucanian eggplant should be phenolic acids, known to be the most abundant active molecules in *S. aethiopicum* species [12]. It was indeed demonstrated that chlorogenic acid carries out its anti-inflammatory activity through the downregulation of the transcriptional factor NF-κB [35]. The anti-inflammatory effects evidenced in this study were also in line with that demonstrated by an in vivo experimental rats model of inflammation treated with the African *S. aethiopicum* [44].

## 3. Materials and Methods

### 3.1. Chemicals and Reagents

Absolute ethanol, Dulbecco’s Modified Eagle Medium (DMEM), dimethyl sulfoxide (DMSO), [3-(4,5-dimethyl-2-thiazolyl)-2,5-diphenyl-2H-tetrazolium bromide] (MTT) Oil Red O, and 2′,7′-dichlorodihydrofluorescein diacetate (DCFH-DA) were purchased from Sigma Aldrich S.p.A. (Milan, Italy). Trypsin-EDTA solution, fetal bovine serum (FBS), glutamine, penicillin–streptomycin, and phosphate saline buffer (PBS) were purchased from Euroclone (Milan, Italy). Reagents used for qRT-PCR were purchased from Euroclone (Milan, Italy).

### 3.2. Cell Culture

Human hepatocellular carcinoma cell line (HepG2) and human colorectal adenocarcinoma cell line (Caco-2) were cultured in DMEM (supplemented with 10% fetal bovine serum, 2 mM glutamine, 100 U/mL penicillin, and 100 μg/mL streptomycin) and maintained at 37 °C in a humidified atmosphere containing 5% CO_2_.

### 3.3. Cell Viability

Cell viability was evaluated by MTT assay as described by Faraone et al. 2022 [12]. Briefly, Caco-2 cells were seeded (1.5 × 10^3^ cells/well) in a 96-well plate and treated with different doses (5–400 μg/mL) of the Lucanian *S. aethiopicum* peel extract for 24 h. 

### 3.4. Determination of Total Lipid Accumulation in HepG2 Cell Line by Oil Red O Staining

HepG2 cells were seeded in 24 wells (1.8 × 10^5^ cells/well) and treated with different doses of the Lucanian *S. aethiopicum* peel extract (200–100 μg/mL) and oleic acid (0.3 mM OA/BSA solution) for 24 h. Subsequently, cells were washed twice with phosphate buffer saline (PBS), and they were fixed with 4% paraformaldehyde for 30 min. Then cells were stained with Oil red O working solution for 20 min at room temperature. The Oil Red O stock solution was prepared by dissolving 0.35 g of Oil Red O (Sigma Aldrich, Milan, Italy) in 100 mL of isopropanol by gentle heating and then cooled and filtered through a 0.45 μm filter. The working solution was prepared by diluting three parts of the stock solution in two parts of water (stock solution: water; 3:2 *v*/*v*) [45]. After several washings, cells were observed using fluorescence microscopy Floid CellTM Imaging Station (Thermo Fisher Scientific, Waltham, MA, USA).

### 3.5. Intracellular ROS Determination

HepG2 cells were plated at a density of 1.8 × 10^5^ cells/well in 24-well plate and treated with different concentrations of Lucanian *S. aethiopicum* peel extract (200–100 μg/mL) and oleic acid (0.3 mM) for 24 h. Then, ROS was determined by BD FACSCanto II flow cytometry (Becton Dickinson, Sunnyvale, CA, USA) as described by Sinisgalli et al. 2020 [46].

### 3.6. Quantitative qRT-PCR

HepG2 or Caco-2 were seeded in 6-well plates (8.5 × 10^5^ cells/well) and treated with different doses of Lucanian *S. aethiopicum* peel extract (200–100 μg/mL) and oleic acid (OA 0.3 Mm) for 24 h. RNA extraction and qRT-PCR were performed, as reported by Armentano et al. 2018 [47]. The specific primers for qRT-PCR are listed in Table 1.

### 3.7. Statistical Analysis

Data were expressed as mean ± standard deviation (Mean ± SD). Statistical analysis was performed using GraphPad Prism 5 Software, Inc. (San Diego, CA, USA), and *p*-values ≤ 0.05 were considered statistically significant.

## 4. Conclusions

This study demonstrated the promising anti-obesogenic activity of the Lucanian *S. aethiopicum* peel extract since it counteracted fat accumulation by promoting lipolysis and lipid elimination instead of lipogenesis. Indeed, the extract could regulate the expression of key molecular targets involved in lipogenesis by down-regulating SREBP-1 and HMGCR and fatty acid *β*-oxidation by up-regulating PPARα, CPT1A, and UCP2. Furthermore, the extract also improved oxidative stress and inflammatory status caused by prolonged exposure to an excess of fat by regulating the molecular pathway coordinated by Nrf2 and Nf-κB and reducing ER stress activation. Considering these results, it is possible to view the Lucanian *S. aethiopicum* peel extract as a promising source of active molecules usable for preventing obesity and managing NAFLD.

## Figures and Tables

**Figure 1 plants-12-02859-f001:**
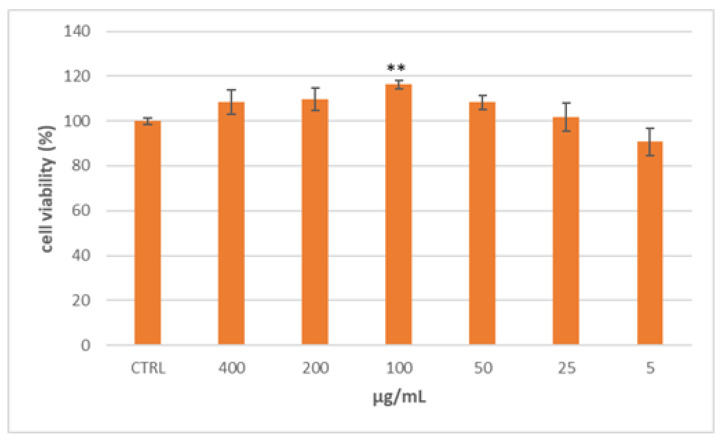
Cell viability of Caco-2 cells treated for 24 h with different concentrations of the Lucanian *S. aethiopicum* peel extract evaluated by MTT assay. Data are expressed as the mean ± SD of three independent experiments (*n* = 3). ** *p* < 0.01 vs. untreated cells (CTRL).

**Figure 2 plants-12-02859-f002:**
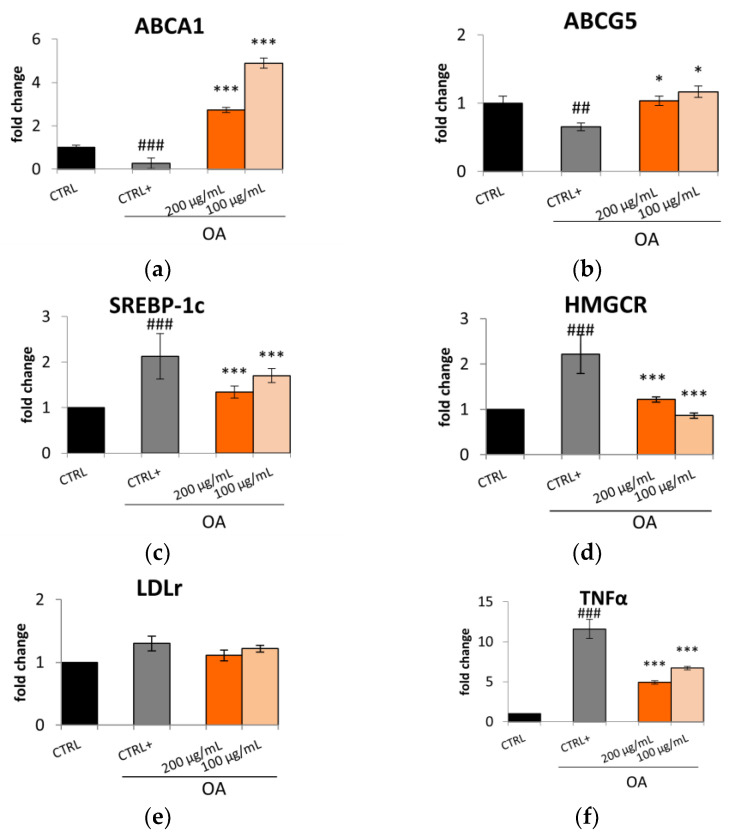
Effect of Lucanian *S. aethiopicum* peel extract (200 and 100 μg/mL) on the gene expression of (**a**) ATP-binding cassette transporter A1 (ABCA1), (**b**) ATP-binding cassette transporter G5 (ABCG5), (**c**) sterol regulatory element-binding protein 1c (SREBP-1c), (**d**) 3-hydroxy-3-methyl-glutaryl-coenzyme A reductase (HMGCR), (**e**) Low-density lipoprotein receptor (LDLr), (**f**) tumor necrosis factor α (TNFα), (**g**) interleukin-6 (IL-6) by real-time qPCR and normalized with the housekeeping gene, *β*-actin, in oleic acid (OA 0.3 mM)-treated Caco-2 cell line. Data are expressed as mean ± standard deviation of three independent experiments (*n* = 3). ## *p* < 0.01, ### *p* < 0.001 vs. untreated cells (CTRL); *** *p* < 0.001, * *p* < 0.05 vs. OA-treated cells (CTRL+).

**Figure 3 plants-12-02859-f003:**
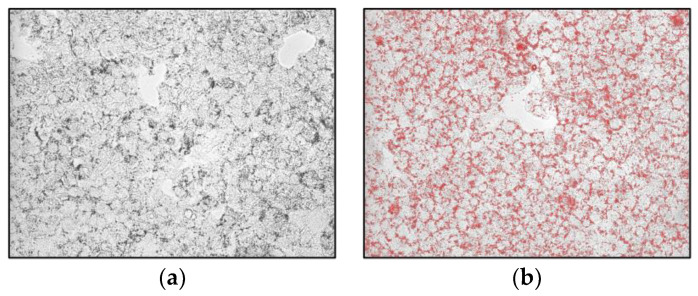
Effects of the Lucanian *S. aethiopicum* peel extract on lipid accumulation in HepG2 cells stimulated with oleic acid (OA) measured by staining with Oil-red-O. (**a**) untreated cells; (**b**) OA 0.3 mM; (**c**) OA + *S. aethiopicum* peel extract 200 μg/mL; (**d**) OA + *S. aethiopicum* peel extract 100 μg/mL.

**Figure 4 plants-12-02859-f004:**
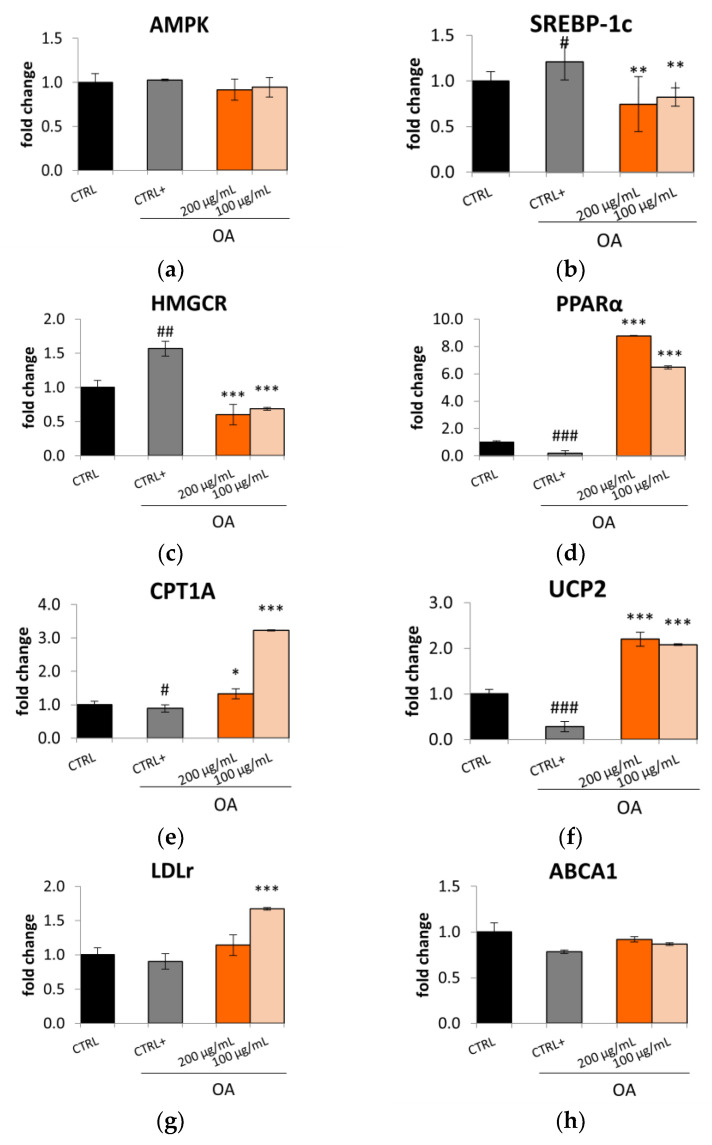
Effect of the Lucanian *S. aethiopicum* peel extract (200 and 100 μg/mL) on the gene expression of (**a**) AMP-activated protein-kinase (**b**) Sterol Regulatory Element-binding Protein-1 (SREBP-1), (**c**) 3-hydroxy-3-methyl-glutaryl-coenzyme A reductase (HMGCR), (**d**) Peroxisome proliferator-activated receptor alpha (PPARα), (**e**) carnitine palmitoyltrasferase 1A (CPT1A), (**f**) uncopling protein 2 (UCP-2), (**g**) Low-density lipoprotein receptor (LDLR), (**h**) ATP-binding cassette transporter A1 (ABCA1), analyzed by real-time q-PCR and normalized with the housekeeping gene, β-actin, in oleic acid (OA 0.3 mM)-treated HepG2 cell line. Data are expressed as mean ± standard deviation of three independent experiments (*n* = 3). # *p* < 0.05, ## *p* < 0.01, ### *p* < 0.001 vs. untreated cells (CTRL); * *p* < 0.05, ** *p* < 0.01, *** *p* < 0.001 vs. oleic acid-treated cells (CTRL+).

**Figure 5 plants-12-02859-f005:**
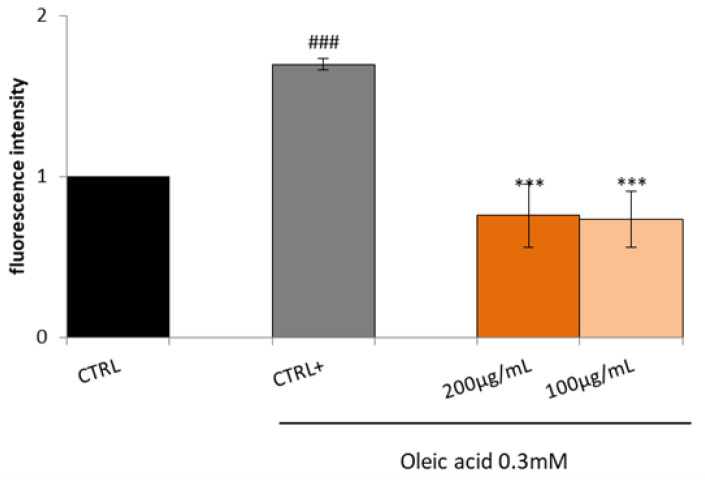
Effect of Lucanian *S. aethiopicum* peel extract on oleic acid (OA)-induced intracellular reactive oxygen species (ROS) generation. HepG2 cells were treated with different concentrations of *S. aethiopicum* peel extract (200–100 μg/mL) and oleic acid (OA 0.3mM) (CTRL+) for 24 h. ROS generation was measured by DCFH-DA staining by flow cytometry analysis. Data are expressed as the mean ± SD of three independent experiments (*n* = 3). ### *p* < 0.001 vs. untreated cells (CTRL), *** *p* < 0.001 vs. OA-treated cells (CTRL+).

**Figure 6 plants-12-02859-f006:**
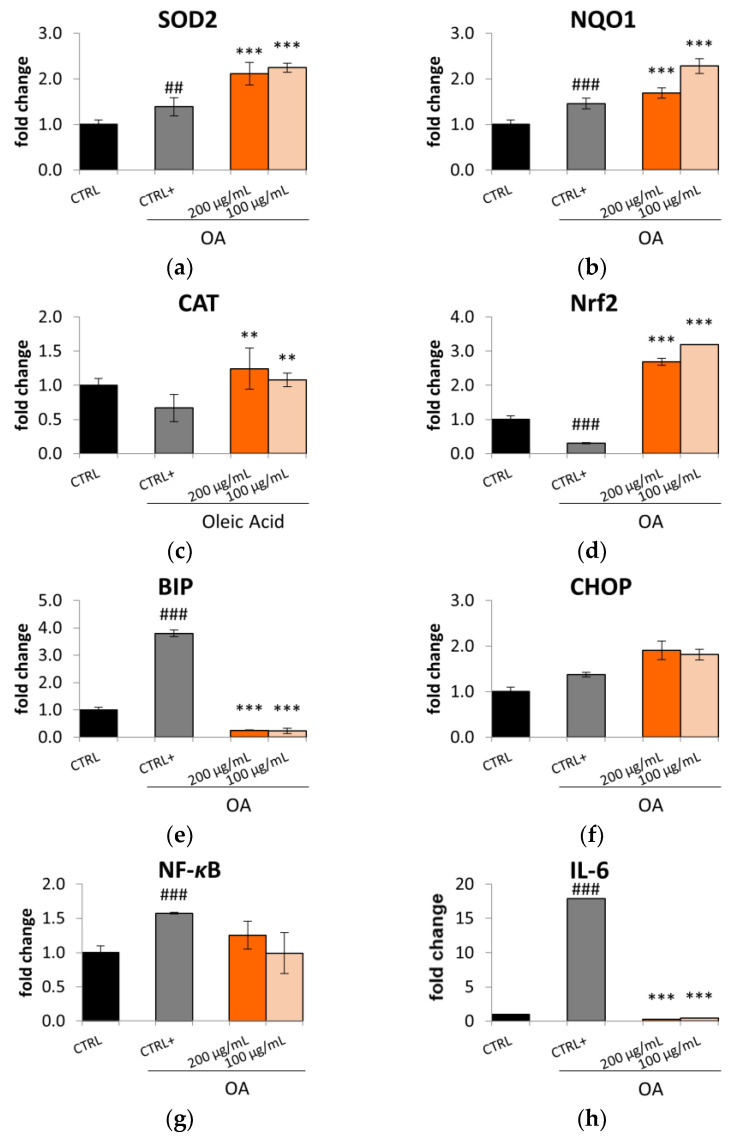
Effect the Lucanian *S. aethiopicum* peel extract (200 and 100 μg/mL) on the gene expression of (**a**) superoxide dismutase 2 (SOD2), (**b**) NADPH quinone dehydrogenase 1 (NQO1), (**c**) catalase (CAT), (**d**) nuclear factor erythroid 2-related factor (Nrf2), (**e**) binding immunoglobulin protein (BIP), (**f**) (CHOP), (**g**) factor kappa light chain enhancer of activated B cells (NF-κB), (**h**) interleukin-6 (IL-6), analyzed by real-time q-PCR and normalized with the housekeeping gene, *β*-actin, in oleic acid (OA 0.3 mM) treated HepG2 cells. Data are expressed as mean ± standard deviation of three independent experiments (*n* = 3). ## *p*< 0.01, ### *p* < 0.001 vs. untreated cells (CTRL), *** *p* < 0.01, ** *p* < 0.001 vs. OA-treated cells (CTRL+).

**Table 1 plants-12-02859-t001:** Reverse and forward primers used for qRT-PCR.

Gene	Forward Primer	Reverse Primer
*β*-actin	5′-CCTGGCACCCAGCACAAT-3′	5′-GCCGATCCACACGGAGTACT-3′
ABCG5	5′-GCTCCAGGATCCTAAGG-3′	5′-GAAAAAGCTCAGAACGG-3′
ABCA1	5′-GGACATGCACAAGGTCCTGA-3′	5′-CAGAAAATCCTGGAGCTTCAAA-3′
BIP	5’-GTTTGCTGATAATTGGTTGAACA-3’	5’-GAATCGCCTGACACCTGAAGA-3’
CAT	5′-ATACCTGTGAACTGTCCCTACCG-3′	5′-GTTGAATCTCCGCACTTCTCCAG-3′
CHOP	5’-TCT CCT TCA TGC GCT GCT TTC-3′	5’-GTA CCT ATG TTT CAC CTC CTG-3′
NQO1	5′-GGTGGTGGAGTCGGACCTCTA-3′	5′-AGGGTCCTTCAGTTTACCTGTGAT-3′
Nrf2	5′-AACTACTCCCAGGTTGCCCA-3′	5′-CATTGTCATCTACAAACGGGAA-3′
SOD2	5′-CCGACCTGCCCTACGACTAC-3′	5′-AACGCCTCCTGGTACTTCTCC-3′
AMPK	5’-TTCAAAAGGCTAATCACAGAAG-3’	5’-TTCAGGAAGATTGTATGCAGG-3’
CPT1A	5’-TTTGCAGTGCCCATCCTCCG-3′	5’-ACAGGTGGTTTGACAAGTCG-3’
HMGCoA (HMGCR)	5’-GCACCTTTCTCTGCATTC C-3’	5’-CGAGAAAGAAAAGTTGAGGT-3’
PPARα	5’-GAATCGCGTTGTGTGACATG-3’	5’-AGG CTGCAAGGGCTTCTTTC-3’
SREBP-1c	5’-TCCTTCAGAGATTTGCTTTTG-3’	5’-TGAGGCAAAGCTGAATAAATC-3’
UCP2	5’-TTACGAGCAACATTGGGAGAG-3′	5’-GCTGGAGGTGGTCGGAGATA-3’
LDLr	5′-AGTGCGGGACCAACGAA-3′	5′-ATGGAGCCCACAGCCTT-3′
TNFα	5’-CACGATCAGGAAGGAGAAGA-3’	5’-TCTGGCCCAGGCAGTCAGAT-3’
IL-6	5’-CTACTCTCAAATCTGTTCTGG-3’	5’-GGATTCAATGAGGAGACTTG-3’

## Data Availability

Not applicable.

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
