# Peer review of "Solanum aethiopicum L. from the Basilicata Region Prevents Lipid Absorption, Fat Accumulation, Oxidative Stress, and Inflammation in OA-Treated HepG2 and Caco-2 Cell Lines"

_plants, 2023, doi:10.3390/plants12152859_

Round 1

Reviewer 1 Report

The article under evaluation presents the effect of S. aethiopicum L. peel extract on some genes involved in lipid absorption, fat accumulation, oxidative stress, and inflammation using Hep G2 and Caco2 cell lines. The current paper - beautifully presented - is a continuation of the article in Pharmaceutics (2022), with some partially overlapping results.

There are also certain techniques/methods that, although presented in the "Materials and Methods" section, are repeated in detail (eg, line 90).

I did not understand why a basic technique in a laboratory was presented in such detail. And in the end, I did not see the conclusion for which 24 hours and not 48 hours was taken as a reference.

In another order, looking at the literature, it does not seem that the association of Solanum aethiopicum L. and the Caco2 cell line is unique. There may be other associations (eg, the choice of tested genes).

Author Response

Response to Reviewer 1 Comments

The article under evaluation presents the effect of S. aethiopicum L. peel extract on some genes involved in lipid absorption, fat accumulation, oxidative stress, and inflammation using Hep G2 and Caco2 cell lines. The current paper - beautifully presented - is a continuation of the article in Pharmaceutics (2022), with some partially overlapping results.

There are also certain techniques/methods that, although presented in the "Materials and Methods" section, are repeated in detail (eg, line 90).

I did not understand why a basic technique in a laboratory was presented in such detail. And in the end, I did not see the conclusion for which 24 hours and not 48 hours was taken as a reference.

Response: We thank you for the suggestion. We have removed the description of the MTT essay from the discussion and results section. About the Caco-2 time of treatment, we tested the extract at 24 and 48 hours without observing any toxicity; the choice for considering the 24h is related to the observation for which many investigations in the literature evaluating plant extract cytotoxicity on Caco-2 until 24 h (https://doi.org/ 10.22034/APJCP.2018.19.6.1697 or https://doi.org/10.1016/j.fct.2014.12.016). Furthermore, we have decided to go ahead within 24 hours as this time range is usually used for this type of study. https://www.ncbi.nlm.nih.gov/pmc/articles/PMC9914695/ or https://www.mdpi.com/1420-3049/27/17/5368. In any case, the reference to 48h was removed to avoid misunderstandings.

In another order, looking at the literature, it does not seem that the association of Solanum aethiopicum L. and the Caco2 cell line is unique. There may be other associations (eg, the choice of tested genes).

Response: We thank you for the observations. There is no unique association between Solanum aethiopicum L. and the Caco2 cell line; as explained in the text (lines 79 – 89), the choice of this cellular model is based on the necessity to evaluate the effect of the extract on intestinal lipid absorption.

Reviewer 2 Report

This manuscript descripted the investigation about the effect of the Lucanian S. aethiopicum peel extract on lipid absorption, lipid accumulation, fatty acid metabolism, oxidative stress, and inflammation on in vitro cells, results suggested the extract could prevents lipid absorption, fat accumulation, oxidative stress, and inflammation in OA-treated HepG2 and Caco-2 cell lines. It’s a relative systematic work and seems interesting to the audiences, however, it still need major revision because there are some bugs in this manuscript.

1. Most important, the authors had suggested that the phenolic acids should be the active ingredients, why not make an enriched separation and performed some experiments with the enriched phenolic acids? Bioassay-guided investigation on the definitely active ingredients might be more convincible. I think only the results of extract could not be considered in the journal of Plants.

2. Some revisions in the manuscript are needed.

  (1) line 106, Caco-2 cells were treated for 24h with oleic acid (OA) 0.3 mM, the authors should give an explanation, why choose this concentration of 0.3 mM? The author can also provide literature.

  (2) line 172, figure captions of Figure 2, the authors should give a detail illustration about CTRL, CTRL+, a good paper should distinctly present the results by some data in some corresponding figures. So, the figure captions are quite important. The same revisions are needed for all other figure captions. Besides, the concentration of OA in line 177 should be 0.3 mM, not 0.3 Mm.

  (3) Keywords, too many keywords in line 25. Generally, five keywords are enough.

Author Response

Response to Reviewer 2 Comments

This manuscript descripted the investigation about the effect of the Lucanian S. aethiopicum peel extract on lipid absorption, lipid accumulation, fatty acid metabolism, oxidative stress, and inflammation on in vitro cells, results suggested the extract could prevents lipid absorption, fat accumulation, oxidative stress, and inflammation in OA-treated HepG2 and Caco-2 cell lines. It’s a relative systematic work and seems interesting to the audiences, however, it still need major revision because there are some bugs in this manuscript.

  1. Most important, the authors had suggested that the phenolic acids should be the active ingredients, why not make an enriched separation and performed some experiments with the enriched phenolic acids? Bioassay-guided investigation on the definitely active ingredients might be more convincible. I think only the results of extract could not be considered in the journal of Plants.

Response: We thank you for the observations. We will consider the study concerning extracts enriched with phenolic acids. As for the current study, we think it is appropriate for the journal since there are other published articles concerning the biological activity of the crude extract: https://doi.org/10.3390/plants12132502 , https://doi.org/10.3390/plants12132533 , https://doi.org/10.3390/plants12132396

  1. Some revisions in the manuscript are needed.

  (1) line 106, Caco-2 cells were treated for 24h with oleic acid (OA) 0.3 mM, the authors should give an explanation, why choose this concentration of 0.3 mM? The author can also provide literature.

Response: the oleic acid concentration used was chosen based on internal laboratory tests

  (2) line 172, figure captions of Figure 2, the authors should give a detail illustration about CTRL, CTRL+, a good paper should distinctly present the results by some data in some corresponding figures. So, the figure captions are quite important. The same revisions are needed for all other figure captions. Besides, the concentration of OA in line 177 should be 0.3 mM, not 0.3 Mm.

Response: We thank you for the observations; where missing, we have added the mining of CTRL. Regarding the mistake in line 177, we have corrected it.

  (3) Keywords, too many keywords in line 25. Generally, five keywords are enough.

Response: We thank you for the suggestion, but following the journal guideline, Three to ten pertinent keywords are allowed; in our case, are eight.

Round 2

Reviewer 2 Report

lt can be accepted.